# Significance of LIF/LIFR Signaling in the Progression of Obesity-Driven Triple-Negative Breast Cancer

**DOI:** 10.3390/cancers16213630

**Published:** 2024-10-28

**Authors:** Lois Randolph, Jaitri Joshi, Alondra Lee Rodriguez Sanchez, Uday P. Pratap, Rahul Gopalam, Yidong Chen, Zhao Lai, Bindu Santhamma, Edward R. Kost, Hareesh B. Nair, Ratna K. Vadlamudi, Panneerdoss Subbarayalu, Suryavathi Viswanadhapalli

**Affiliations:** 1Department of Obstetrics and Gynecology, University of Texas Health San Antonio, San Antonio, TX 78229, USA; randolphl@uthscsa.edu (L.R.); rodriguezsaa@uthscsa.edu (A.L.R.S.); pratap@uthscsa.edu (U.P.P.); gopalam.rahul@gmail.com (R.G.); kost@uthscsa.edu (E.R.K.); vadlamudi@uthscsa.edu (R.K.V.); 2Department of Biochemistry, University of Wisconsin-Madison, Madison, WI 53706, USA; jjoshi6@wisc.edu; 3Greehey Children’s Cancer Research Institute, University of Texas Health San Antonio, San Antonio, TX 78229, USA; cheny8@uthscsa.edu (Y.C.); laiz@uthscsa.edu (Z.L.); 4Department of Population Health Sciences, University of Texas Health San Antonio, San Antonio, TX 78229, USA; 5Department of Molecular Medicine, University of Texas Health San Antonio, San Antonio, TX 78229, USA; 6Evestra, Inc., San Antonio, TX 78245, USA; bsanthamma@evestra.com (B.S.); hnair@evestra.com (H.B.N.); 7Mays Cancer Canter, University of Texas Health San Antonio, San Antonio, TX 78229, USA; 8Audie L. Murphy Division, South Texas Veterans Health Care System, San Antonio, TX 78229, USA

**Keywords:** EC359, obesity, TNBC, LIF/LIFR signaling, RNA-seq

## Abstract

The obesity epidemic in the USA is increasing the risk of aggressive triple-negative breast cancer (TNBC) in obese women. This study investigated the potential impact of obesity on the advancement of TNBC by amplifying leukemia inhibitory factor receptor (LIFR) signaling. We tested the effects of LIFR inhibition using EC359 on TNBC cells in obesity conditions. The obesity environment increased TNBC cell growth and invasion, and treatment with EC359 effectively reduced these effects. Using TNBC cells, patient-derived organoids, and mouse models, we showed that EC359 can inhibit obesity-linked TNBC progression. Collectively, our results suggest targeting LIFR signaling as a potential treatment for obesity-related TNBC.

## 1. Introduction

Triple-negative breast cancer (TNBC), a subtype of breast cancer that lacks estrogen, progesterone, and HER2 receptors, accounts for 15–20% of BC cases and contributes disproportionately to BC mortality due to a lack of targeted therapies [1,2]. The incidence of TNBC is higher in obese women and obese TNBC patients exhibit a larger tumor size, higher grade, and stage [3,4]. Obesity is an independent risk factor that increases the risk of TNBC progression, therapy resistance, and metastases [5,6]. The currently available targeted therapies are only useful in a subset of TNBC patients with mutations (such as BRCA1). While chemotherapy has proved effective in some TNBC patients, therapy resistance and relapse are very common and thus limit the utility of chemotherapy. New targeted therapies are needed for the efficient management of TNBC.

Obesity-driven signaling functions as a potent inducer of the leukemia inhibitory factor (LIF), the most pleiotropic member of the interleuki-6 family of cytokines [7]. The LIFR-gp130 complex mediates LIFR signaling [8]. LIFR does not have inherent tyrosine kinase activity. LIFR and gp130 constitutively connect with the JAK-Tyk family of cytoplasmic tyrosine kinases, and LIF binding to the LIFR complex activates JAK, STAT, MAPK, AKT, and mTOR [8,9,10]. LIF and its receptor LIFR are commonly over-expressed in multiple solid tumors [11] and tumors upregulate LIF/LIFR signaling via autocrine and paracrine mechanisms [12,13,14]. LIF/LIFR signaling plays a key role in tumor growth, progression, metastasis, stemness, and therapy resistance [10,15,16,17]. LIF/LIFR activation occurs more prominently in TNBC compared to estrogen receptor positive (ER^+^) BC, and high circulating LIF levels correlate with tumor recurrence and contribute to chemoresistance [14,18]. However, the role of LIF/LIFR signaling in the progression of obesity-driven TNBC remains elusive.

Recent studies suggested that the LIF/LIFR axis represents a potential therapeutic target. A LIF blockade or genetic LIFR deletion slow tumor progression and augment the efficacy of chemotherapy [16]. Another study confirmed that the blockade of LIF improves the therapeutic outcome [17]. With our expertise in developing novel small-molecule inhibitors, we have rationally designed and synthesized a small organic molecule that can emulate the LIF–LIFR binding site and functions as a LIFR inhibitor (EC359) [19]. EC359 represents a first-in-class drug to inhibit LIFR oncogenic functions. In this study, we tested the hypothesis that obesity enhances LIFR signaling in TNBC, contributing to cancer progression, and tested the therapeutic potential of LIFR inhibition using the small-molecule inhibitor EC359.

## 2. Materials and Methods

### 2.1. Cell Culture and Reagents

TNBC cell lines (MDA-MB-231, BT-549, SUM159, and 4T1) were purchased from ATCC and were maintained using ATCC-recommended medium. Human primary mature adipocytes (ADP) were isolated from subcutaneous fat samples obtained from UT Health ObGyn tissue core that obtained these tissues with informed consent using approved IRB protocol. In this study, we used six distinct adipose tissue samples, labeled ADP-1 through ADP-6. Conditioned medium collected to form these tissue samples was named as ADP-CM-1-to-6. Each sample was obtained from a different individual with a Body Mass Index (BMI) exceeding 30, ensuring that the samples were representative of individuals with obesity for our analysis. Adipocyte-conditioned medium (ADP-CM) was prepared by incubating 2 mL of packed primary mature adipocytes in 6 mL of serum-free DMEM medium in a 100 cm^2^ petri dish for 48 h. In all in vitro experiments, basal DMEM medium was used as serum-free medium (SFM) and was combined with 50% (*v*/*v*) ADP-CM. The synthesis of LIFR inhibitor EC359 was earlier described [19].

### 2.2. Co-Culture and Treatment Conditions

Mature adipocytes were cultured in DMEM supplemented with 0.5% FBS. For the co-culture experiment, 1 mL of packed primary mature adipocytes was added to an insert in a Boyden chamber, while MDA-MB-231 cells were seeded in the bottom well of a 6-well plate. Control wells had the insert without mature adipocytes. The co-cultures were performed in triplicate for 24 h with or without EC359 (100 nM) treatment. Following this, MDA-MB-231 cells were lysed to isolate proteins for Western blot analysis.

### 2.3. Cell Viability, Colony Formation, and Invasion Assays

For cell viability assays, TNBC cells were seeded at a density of 1 × 10^3^ cells per well in a 96-well plate and incubated with ADP-CM, with or without EC359 at concentrations of 100 and 50 nM. After five days of drug treatment, cell viability was assessed using the MTT assay. Colony formation was evaluated by seeding cells at low density (500 cells per well) and culturing them with ADP-CM and EC359 (50 nM), the medium was replaced with regular cell culture medium, and the cells were cultured for an additional 14 days to allow colonies to form. The colonies developed during long-term culture were then counted. Invasion assays were performed using Matrigel-coated Transwell chambers as described [19]. For these assays, 100,000 cells were plated in the upper chamber with or without ADP-CM and with or without EC359 (100 nM), while regular cell culture medium was added to the lower chamber. After 22 h of incubation, the cells that had invaded through the Matrigel were stained with 0.5% crystal violet. The number of invaded cells was then counted per microscopic field.

### 2.4. Generation of LIFR-Knockdown Cells

MDA-MB-231/STAT3/luc cells were transfected using Lipofectamine RNAiMAX Reagent (Thermo Fisher, Waltham, MA, USA; cat# 13778150) following the manufacturer’s instructions. Cells were transfected with either scrambled siRNA (siRNA Universal Negative Control #1) or LIFR-targeting siRNAs (siLIFR-1 and siLIFR-2) from Sigma-Aldrich (St. Louis, MO, USA). Specifically, siLIFR-1 corresponds to the LIFR siRNA ID SASI_Hs02_00330114, and siLIFR-2 corresponds to SASI_Hs02_00330115. Transfection was allowed to proceed for 72 h, after which the cells were analyzed by STAT3 reporter assay.

### 2.5. Western Blotting and RT-qPCR

RT-qPCR analysis was used to determine mRNA levels and Western blotting was used to evaluate protein expression, using standard protocols [19]. The antibodies used in this study were p-p70S6K(T389), p70S6K, p-Akt(S473), Akt, p-mTOR(S2448), mTOR, p-S6(S235/236), S6, p-ERK1/2, ERK, p-STAT3(Y705), and STAT3, which were purchased from Cell Signaling Technology (Beverly, MA, USA), the LIF and LIFR were acquired from Santa Cruz Biotechnology (Dallas, TX, USA), and anti-β-actin from Sigma (St. Louis, MO, USA). Using a CFX96 Real-Time PCR instrument and a High-Capacity cDNA Reverse Transcription Kit and PowerUp SYBR Green master mix (Applied Biosystems, Foster City, CA, USA), RT-qPCR was performed. Appendix A has primer sequences and Appendix A have the original Western blots.

### 2.6. Reporter Gene Assays

STAT3 reporter gene assays were conducted as described previously [19]. Briefly, STAT3-luc reporter cells were deprived of serum for 24 h, followed by treatment with either ADP-CM alone or ADP-CM combined with EC359 for another 24 h. The activity of the reporter was then measured using a luminometer and the luciferase reporter assay system (Promega, Madison, WI, USA).

### 2.7. Obesity Induction

For diet-induced obesity studies, Rodent diet with 60 kcal% fat was used for high-fat-diet (HFD) groups (Research Diets, Inc., #D12492 New Brunswick, NJ, USA) and for low-fat-diet (LFD) groups, rodent diet with 10 kcal% fat was used (Research Diets, Inc. #D12450J New Brunswick, NJ, USA). Body weight was monitored weekly twice.

### 2.8. Xenograft and Patient-Derived Organoid Models

TNBC xenografts were established by injecting MDA-MB-231 into the mammary fat pad of immunocompromised SCID mice, with or without co-implantation of adipocytes or with low-fat-diet (LFD)/high-fat-diet (HFD) feeding. All animal experiments were conducted using approved UT Health San Antonio IACUC protocol and guidelines. For adiposity-driven studies, MDA-MB-231 cells and mature ADP cells (1 × 10^6^) were mixed with equal volume of growth factor-reduced Matrigel and injected orthotopically into 8-week-old female SCID mice (n = 7 tumors/group). For diet-induced obesity studies, female SCID mice (n = 5/group) were fed with HFD or LFD. After 8 weeks, mice were implanted with MDA-MB-231 cells (1 × 10^6^ cells in 100 μL), mixed with an equal volume of growth-factor-reduced Matrigel, in the mammary fat pads. After tumor establishment, mice were randomized into control and treatment groups. The control group received vehicle (0.3% hydroxy propyl cellulose) and the treatment group received EC359 (5 mg/kg/ip/5 days/week). All mice were monitored daily for adverse toxic effects. Tumor growth was measured with a caliper at 3–4-day intervals, and volume was calculated using a modified ellipsoidal formula: tumor volume = 1/2(L × W^2^), where L is the longitudinal diameter and W is the transverse diameter. At the end of the experiment, mice were euthanized, and tumors were excised and weighed.

For patient-derived organoid ex vivo models, mice bearing TNBC PDX (TM00096) tumors were purchased from Jackson laboratory (Bar Harbor, ME, USA). When tumors reached ~750 mm^3^ size, they were dissected into small pieces, digested with collagenase, and were cultured as previously described [20]. For cell viability assays, 5 × 10^3^ cells in a 10 μL droplet were seeded into each well of a 96-well plate. Organoid cultures were treated with EC359 or vehicle control, with or without ADP-CM, in triplicate. After 7 days of treatment, cell viability was assessed using the Promega^®^ CellTiter-Glo^®^ 3D-Superior Cell Viability Assay reagent according to the manufacturer’s instructions (Promega, Madison, WI, USA). Luminescence intensity was measured using a GloMax^®^ Discover System (Promega, Madison, WI, USA).

### 2.9. Syngeneic Xenograft Model

After 6 weeks of diet initiation with either LFD or HFD, 5 × 10^5^ 4T1 cells were injected orthotopically, in a total volume of 100 μL, into the mammary fat pad of BALB/c mice. All animals continued to be fed their respective diets until euthanasia. Tumor volume was measured using a digital caliper and calculated using a modified ellipsoidal formula: tumor volume = 1/2(L × W2), where L is the longitudinal diameter and W is the transverse diameter. After euthanasia, tumors, fat tissue, spleen, and liver were collected and weighed.

### 2.10. RNA-Seq Analysis

Total RNA was isolated using RNeasy mini kit (Qiagen, Valencia, CA, USA) from 4T1 syngeneic xenograft tissues. The effect of LFD, HFD, and HFD + EC359 treatment on the global transcriptome was determined. The UT Health genome sequencing core facility sequenced RNA. The sequence data were aligned to the mouse genome (UCSC mm9) using TopHat2. HTSeq quantified genes and NCBI RefSeq annotated them. DESeq identified differentially expressed genes [21]. Genes having a fold change larger than 2 and a multiple-test-adjusted *p*-value below 0.01 were used for pathway analysis. GSEA [22] was conducted on 16 October 2023.

### 2.11. Statistical Analysis

Data were analyzed using GraphPad Prism software (version 9.5.1. Dotmatics, Boston, MA, USA). Statistical significance was determined by ANOVA or *t*-test, with *p* < 0.05 considered significant.

## 3. Results

### 3.1. Adipose Conditions Promote TNBC Cell Proliferation and LIFR Downstream Signaling 

TNBC cells (BT-549 and SUM-159) were cultured in adipocyte-conditioned medium (ADP-CM) for 24 h. The expression of LIFR target genes was assessed using RT-qPCR. Under conditions of obese-related adiposity, there was a significant upregulation of LIFR target genes in both BT-549 and SUM-159 cells compared to control conditions (Figure 1A,B). Western blot analysis revealed increased expression or activation of LIFR downstream signaling proteins in TNBC cells treated with ADP-CM compared to untreated cells (Figure 1C, Appendix A). Collectively, these results suggest that ADP-CM enhances LIFR downstream signaling at the protein level, corroborating the gene expression data from RT-qPCR.

### 3.2. LIFR Inhibition with EC359 Suppresses TNBC Cell Growth

TNBC cells were treated with EC359 both with and without the presence of adipose conditions (ADP-CM), and cell viability was assessed using MTT assays. The results demonstrated that EC359 treatment significantly decreased TNBC cell viability compared to untreated cells under adipose (ADP-CM) and non-adipose conditions (Figure 1D, Appendix A). Further, cells treated with ADP-CM alone showed enhanced colony formation, indicating increased cell survival under adipose conditions (Figure 1E,F and Appendix A). Addition of EC359 to ADP-CM significantly reduced the number of colonies formed compared to ADP-CM alone. Collectively, these results suggest that EC359 attenuates adiposity-induced cell survival in TNBC cells (Figure 1E,F and Appendix A).

### 3.3. EC359 Inhibits LIFR Downstream Signaling

To evaluate LIFR downstream signaling, we have initially measured STAT3 reporter activity under adipose (ADP-CM) and non-adipose conditions. The results showed that ADP-CM significantly increased STAT3 reporter activity. The presence of EC359 effectively decreased STAT3 reporter activity both with and without the presence of adipose conditions (ADP-CM), suggesting that EC359 inhibits LIFR signaling pathway activation (Figure 2A, Appendix A). To provide direct evidence that the increase in STAT3 reporter activity is initiated primarily through LIFR signaling, we have knocked down LIFR using two distinct siRNAs. Western blotting confirmed the knock down of LIFR. Interestingly, when we knocked down LIFR using two distinct siRNAs, ADP-CM was not able to increase the STAT3 reporter activity (S4D–F). These results, taken together with the data in Figure 2A, support that ADP-CM-mediated increases in the STAT3 activity occur primarily via LIFR signaling.

ADP-CM treatment also led to the elevated expression of LIFR target genes in both BT-549 and MDA-MB-231 cells (Figure 2B). When EC359 was added, the expression of these target genes was significantly reduced compared to cells treated with ADP-CM alone (Figure 2B). In MDA-MB-231 cells, Western blot analyses revealed that ADP-CM treatment increased the levels of phospho-STAT3, while EC359 treatment reduced these levels, further demonstrating that EC359 inhibits LIFR signaling pathway activation (Figure 2C, Appendix A). We then used coculture assays to test the EC359 effect. MDA-MB-231 cells were co-cultured with adipocytes using a transwell system in the presence or absence of EC359 for 24 h. A Western blot analysis of these cells revealed that co-culturing with adipocytes enhanced LIFR downstream signaling activation. However, EC359 attenuated this activation (Figure 2D, Appendix A). To explore the impact of EC359 on the obesity-induced invasion of TNBC cells, Matrigel invasion assays were performed. EC359 significantly reduced the invasion potential of TNBC cells under ADP-CM conditions (Figure 2E,F). In summary, these results demonstrate that obesity conditions enhance LIFR downstream signaling and increase invasion in TNBC cells, as evidenced by increased STAT3 reporter activity, increased LIFR target gene expression, elevated levels of signaling proteins, and enhanced invasion. EC359 effectively mitigates these effects, highlighting its potential as a therapeutic agent for inhibiting LIFR signaling pathway activation in TNBC cells exposed to adipose conditions.

### 3.4. EC359 Inhibits Obesity-Induced Growth in Organoid and Xenograft Models

EC359 treatment significantly inhibited the growth of TNBC patient-derived organoids cultured under adipose conditions (Appendix A). To test the efficacy of EC359 in vivo, we have used xenograft models. MDA-MB-231 cells, along with mature adipocytes mixed in Matrigel were injected into the mammary fat pad of female SCID mice. Once tumors were established, the mice were randomized into control and EC359 treatment groups. The co-implantation of Adipocytes with TNBC cells enhanced TNBC progression, resulting in larger tumors with an increased volume and weight. EC359 treatment abolished obesity-induced TNBC progression (Figure 3A,B and Appendix A).

We next investigated the effects of an HFD on inducing obesity and subsequent TNBC tumor progression in female SCID mice, followed by treatment with EC359 to determine if it could counteract obesity-induced TNBC progression. The mice were randomized into three groups: LFD, HFD, and HFD + EC359. The LFD group was fed a 10% low-fat diet, while the HFD groups were fed a 60% high-fat diet for eight weeks. Body weights were monitored twice a week. After eight weeks, MDA-MB-231 cells mixed with Matrigel (1.0 × 10^6^ cells in 100 μL) were injected into the mammary fat pads of the mice. Following tumor establishment, mice on the HFD had increased body weights compared to those on the 10% low-fat diet (Figure 3C). HFD significantly enhanced tumor growth, which is attenuated by EC359 treatment (Figure 3D,E and Appendix A).

### 3.5. EC359 Inhibits High-Fat-Diet-Induced Syngeneic Xenograft Tumor Growth

To further confirm the utility of EC359 in blocking the obesity-mediated growth of TNBC cells in vivo, we used 4T1 cells in a syngeneic BALB/c mice model. We first fed BALB/c mice with a LFD and HFD for six weeks. After this period, 4T1 cells were injected orthotopically into the mammary fat pad of both groups. Once tumors formed, the mice were randomized into three groups: (1) LFD, (2) HFD, and (3) HFD + EC359. As shown in Figure 4A–G, mice fed with the HFD exhibited significantly enhanced 4T1 xenograft tumor growth. Additionally, treatment with EC359 significantly attenuated HFD-induced TNBC progression (Figure 4A–G). 

To better understand the mechanisms by which obese conditions stimulate TNBC progression, we employed a global RNA-seq approach. For this experiment, we utilized tumor tissues from models fed with a LFD, HFD, and HFD with EC359 treatment. HFD-induced models significantly altered the expression of 578 genes compared to LFD models, and 325 genes were differentially expressed in the HFD + EC359 group compared to the HFD group. Gene set enrichment analysis (GSEA) revealed that genes regulated by the HFD group positively correlated with cytokine signaling pathways, stem cells, inflammation, TGFβ, metastasis, and epithelial–mesenchymal transition pathways (Figure 5A). These pathways were downregulated with EC359 treatment (Figure 5B). Functional analysis using Gene Ontology (NIH DAVID Bioinformatics tool https://david.ncifcrf.gov/tools.jsp (accessed on 1 August 2024) indicated that the upregulated biological processes in the HFD group included cell migration, proliferation, angiogenesis, cell chemotaxis, cell communication, and cell adhesion (Figure 5C), and these pathways were negatively enriched in the EC359-treated groups (Figure 5D). Among the 325 differentially expressed genes identified in the HFD + EC359 group compared to the HFD group, we refined our focus to those genes uniquely associated with the HFD by filtering out the genes overlapping with the LFD group. This analysis resulted in the identification of 179 differentially expressed genes, as shown in the Venn diagram and Heatmap. Subsequent DAVID analysis revealed that these genes were significantly associated with several key pathways related to cell–cell adhesion, the regulation of T cell differentiation, the apoptotic cleavage of cell adhesion proteins, the degradation of the extracellular matrix, the apoptotic cleavage of cellular proteins, and the canonical Wnt signaling pathway (Appendix A).

## 4. Discussion

Obesity is an independent risk factor [23] that increases the risk of TNBC progression [3,24] and metastases [25]. The prevalence of obesity in the USA has reached epidemic proportions, significantly impacting public health. Women with obesity are two times more likely to develop BC, with a 40% increased risk of cancer recurrence and death from disease [5,26]. This study’s findings demonstrate that the obesity-associated microenvironment promotes TNBC progression through the upregulation of LIFR signaling. Adipose conditions enhance TNBC cell proliferation and invasion, correlating with increased LIFR expression and the activation of STAT3 signaling. Importantly, the inhibition of LIFR with EC359 effectively blocks these oncogenic processes, suggesting that LIFR represents a potential therapeutic target for obesity-driven TNBC.

During the past 20 years, the obesity epidemic is rapidly increasing in USA and obese women are at a higher likelihood of developing TNBC [25]. Several studies implicated the importance of the breast microenvironment on aggressive cancer biology, especially the obese microenvironment [3,27]. However, the underlying mechanism(s) by which obesity contributes to the progression of TNBC remains unclear. The obesity microenvironment modulates cytokines [28] and is suspected to promote TNBC progression [3]. Obesity signals (such as leptin) function as potent inducers of the LIF [29,30], the most pleiotropic member of the interleukin-6 family of cytokines [7]. Our results demonstrated that TNBC cells cultured with ADP-CM showed the significant upregulation of LIFR target genes and increased activation of LIFR downstream signaling proteins. These findings indicate that the adipose environment enhances LIFR signaling, potentially promoting tumor progression in TNBC cells.

Published studies showed that tumors upregulate LIF/LIFR signaling via autocrine and paracrine mechanisms [12,13,14]. TNBC often exhibits the autocrine stimulation of the LIFR axis by altering the expression of ligands. For example, endoplasmic reticulum stress and hypoxia are a few of the hallmarks of TNBC [31] and hypoxia induces LIF expression in human cancer cells [32]. CircRNAs such as circSEPT9 regulate the expression of LIF via sponging miR-637 and activate the LIF/Stat3 signaling pathway involved in the progression of TNBC [33]. LIF promotes the proliferation and metastasis of BC cells, and the overexpression of LIF is commonly associated with poorer relapse free survival in BC patients [10]. Further, the increased expression of alternative LIFR ligands such as OSM, CNTF, and CTF1 was reported in TNBC [15]. Our findings are consistent with previous studies indicating the role of LIFR in cancer biology. Further, this study is the first to establish a direct link between obesity, LIFR signaling, and TNBC progression.

The LIF/LIFR axis is linked to cancer proliferation, immune system protection, chemoresistance, and patient survival [11,34,35]. Considering the significance of LIF/LIFR signaling in cancer, our team recently has created an inhibitor for LIFR, designated as EC359. The specificity of EC359 was determined by a microscale thermophoresis assay and LIFR knockdown cells [19]. EC359 exclusively targets cells that express LIF and LIFR. For example, EC359 shows little efficacy against ER^+^ breast cancer cells, such as MCF7, which express minimal amounts of LIF and LIFR. Conversely, TNBC models like MDA-MB-231 exhibit elevated levels of LIF and LIFR, demonstrating significant responsiveness to EC359. Moreover, the depletion of LIFR in the TNBC model cells significantly diminished the efficacy of EC359 [19]. Toxicity investigations indicated no significant abnormalities in any of the treated animal groups at doses of up to 30 mg/kg and established 30 mg/kg as the maximum tolerated dose (MTD) [36]. Published studies showed the efficacy of EC359 in treating triple-negative breast cancer [19,37], endometrial cancer [20,38], pancreatic cancer [39,40], and renal cancer cells [41]. In this study, EC359 at a dosage of 5 mg/kg (which is substantially lower than the MTD dose) showed significant efficacy in mitigating the obesity-induced progression of TNBC. The use of EC359 to inhibit LIFR offers a novel therapeutic approach for managing obesity-associated TNBC, addressing a critical need for effective treatments in the TNBC population.

LIFR does not have inherent tyrosine kinase activity; LIF binding to the LIFR complex activates Jak-STAT as an immediate effector and MAPK, AKT, and mTOR as secondary effectors [8,9,10]. The LIF-STAT3 axis is implicated in stem cell self-renewal and pluripotency [42]. Our results show the suppression of STAT3 reporter activity and the reduced expression of LIFR target genes by EC359 further confirms its inhibitory effect on LIFR downstream signaling. The ability of EC359 to attenuate LIFR signaling in the presence of ADP-CM, as well as in co-culture systems with adipocytes, indicates its broad applicability of targeting LIFR in disrupting obesity-enhanced signaling pathways in TNBC. The RNA-Seq analysis of xenograft models also confirmed critical pathways modulated by obesity via LIF/LIFR signaling, further elucidating the molecular mechanisms by which obesity influences TNBC progression. Collectively, our in vivo and ex vivo findings provide a strong rationale for considering EC359 in clinical settings to manage obesity-driven TNBC.

## 5. Conclusions

The results of this study have demonstrated the efficacy of EC359 in inhibiting the LIFR pathway and have established a critical function of LIFR signaling in mediating the effects of obesity on TNBC progression. EC359 is effective in reducing the proliferation and survival of TNBC cells driven by obesity in vitro, and it substantially impedes tumor growth in organoid and xenograft models. These results indicate that EC359 has the potential to be a therapeutic agent for the treatment of obesity-driven TNBC.

## Figures and Tables

**Figure 1 cancers-16-03630-f001:**
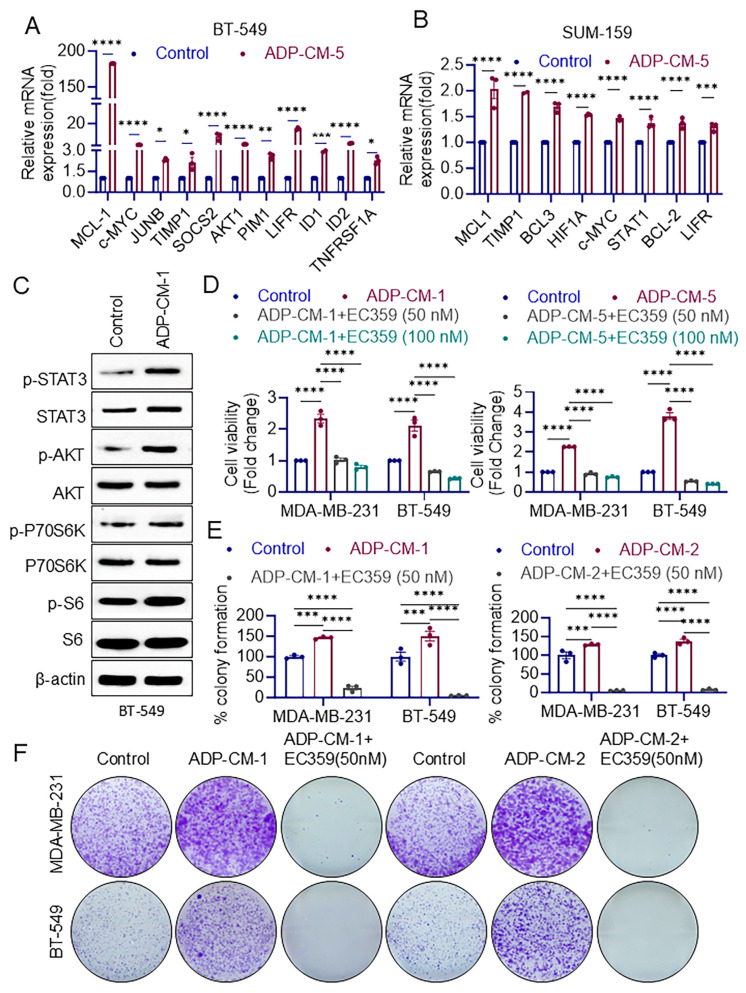
Obese-related adiposity conditions enhance LIFR downstream signaling. TNBC cells, BT-549 (**A**), and SUM-159 (**B**) were cultured with ADP-CM for 24 h, and the expression of LIFR target genes was analyzed by RT-qPCR. (**C**) TNBC cells cultured with ADP-CM for 24 h were analyzed using Western blot analysis to measure LIFR downstream signaling proteins. (**D**) The effects of EC359 treatment against adipose conditions on TNBC cell viability were determined using MTT assays. (**E**) The effects of ADP-CM and ADP-CM + EC359 (50 nM) on adiposity-induced cell survival of TNBC cells was measured using colony formation assays. (**F**) Representative images of colonies were shown. ns, not significant; * *p* < 0.05; ** *p* < 0.01; *** *p* < 0.001; **** *p* < 0.0001.

**Figure 2 cancers-16-03630-f002:**
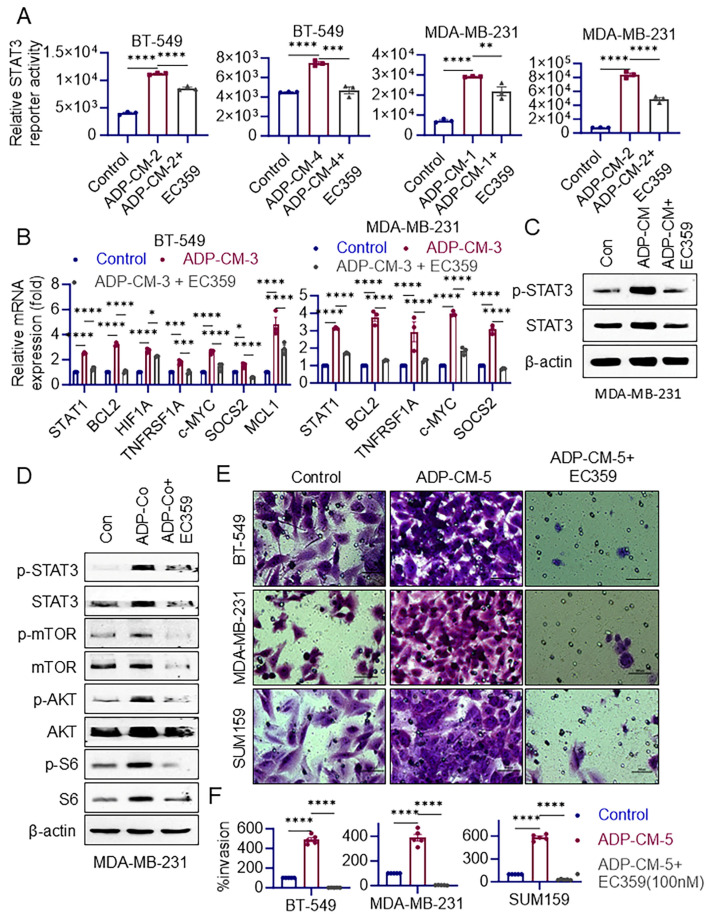
ADP-CM or co-culture of adipocytes enhances LIFR downstream signaling in TNBC and addition of EC359 (100 nM) reduces its activity in vitro. TNBC cells that stably express the STAT3 reporter were used. ADP-CM (**A**) activates STAT3 reporter activity, with EC359 (100 nM) effectively reducing STAT3 activity in ADP-CM conditions. (**B**) BT-549 and MDA-MB-231 cells were cultured with ADP-CM in the presence or absence of EC359 (100 nM) for 24 h, and the expression of LIFR target genes was analyzed by RT-qPCR. (**C**) MDA-MB-231 cells were incubated with or without ADP-CM and EC359 (100 nM) and were analyzed using Western blot analysis to measure LIFR downstream signaling proteins. (**D**) Adipocytes were indirectly co-cultured with MDA-MB-231 cells using a transwell culture system in the presence or absence of EC359 (100 nM) for 24 h. Signaling was profiled by Western blotting. (**E**) Effect of adipose conditions on TNBC cell invasion in the presence or absence of EC359 (100 nM) was determined by Boyden chamber assay and invaded cells were quantified (**F**). * *p* < 0.05; ** *p* < 0.01; *** *p* < 0.001; **** *p* < 0.0001.

**Figure 3 cancers-16-03630-f003:**
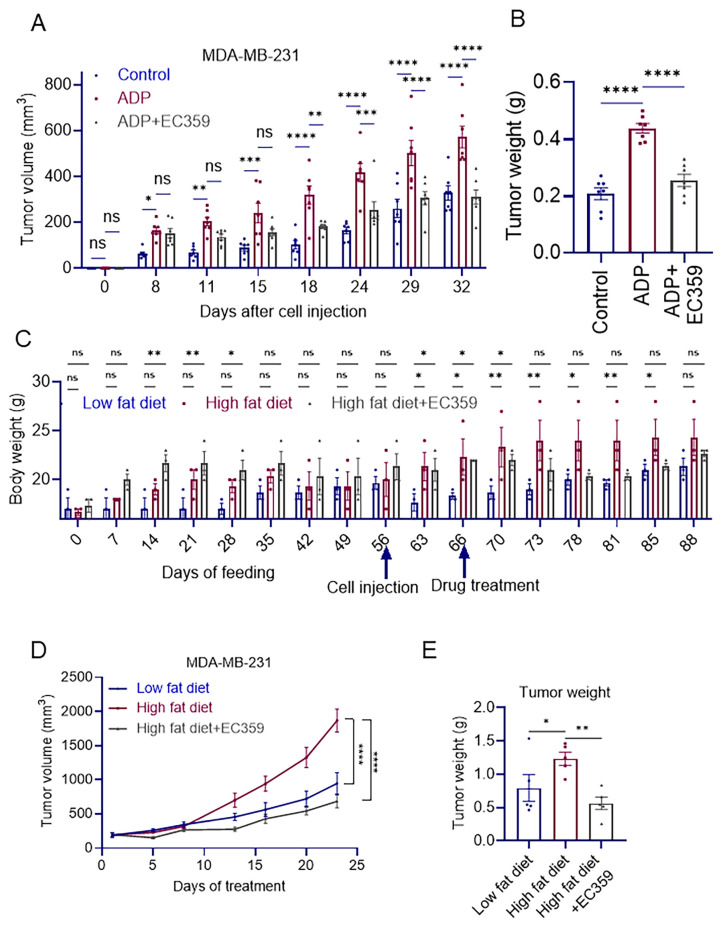
EC359 reduces tumor volume and weight in adiposity and diet-induced TNBC CDX models. Primary human adipose cells were co-implanted with MDA-MB-231 cells via orthotopic injection (n = 7) and subsequently treated with or without EC359 (5 mg/kg, intraperitoneally, 5 days per week). Tumor volumes (**A**) and tumor weights (**B**) are shown in graph. (**C**) Body weights of Low-fat-diet- and high-fat-diet-induced TNBC xenograft models treated with or without EC359 (5 mg/kg/ip/5 days/week) are shown in graph. Tumor volume (**D**) and tumor weights (**E**) are shown. ns, not significant; * *p* < 0.05, ** *p* < 0.01, *** *p* < 0.001, **** *p* < 0.0001.

**Figure 4 cancers-16-03630-f004:**
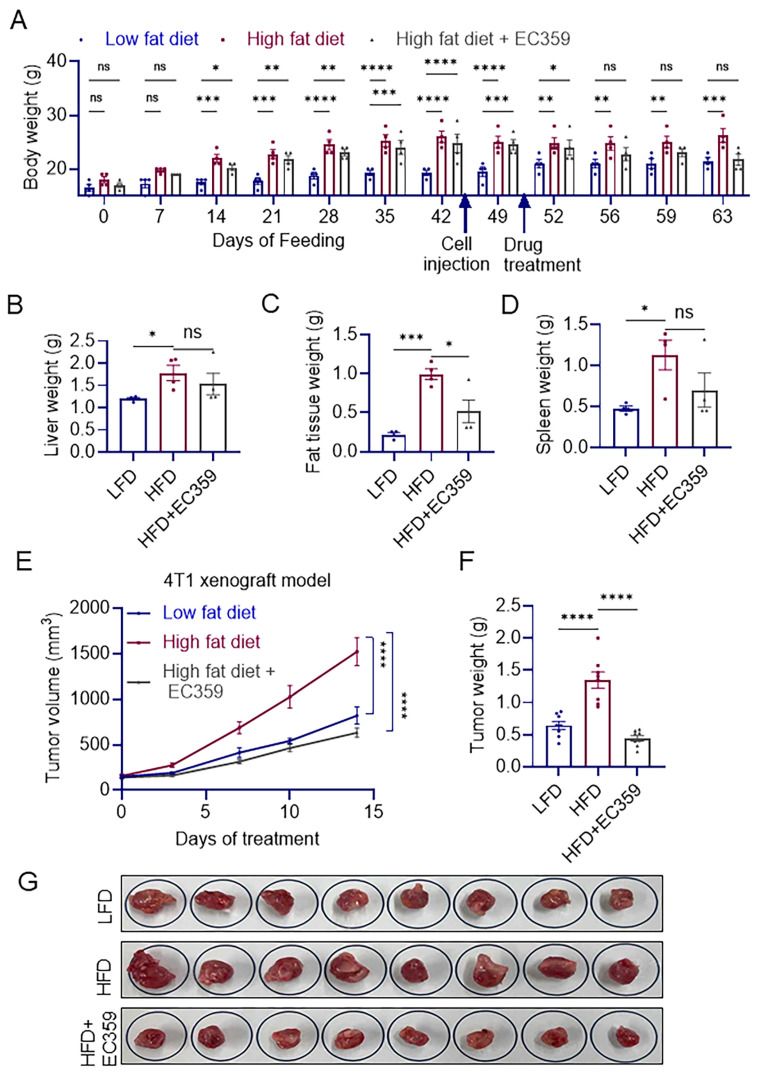
Low-fat diet and high-fat diet 4T1-TNBC syngeneic model. Diet-induced obesity syngeneic models (n = 8) were treated with vehicle or EC359 (5 mg/kg/ip/5 days/week). (**A**) Body weights shown in graph. Bar graphs of liver (**B**), fat tissue (**C**), spleen (**D**), and tumor volume (**E**) were shown. (**F**) Tumor weights of all the 3 groups are shown. (**G**) Representative images of excised tumors from diet-induced syngeneic xenografts treated with or without EC359. ns, not significant; * *p* < 0.05, ** *p* < 0.01, *** *p* < 0.001, **** *p* < 0.0001.

**Figure 5 cancers-16-03630-f005:**
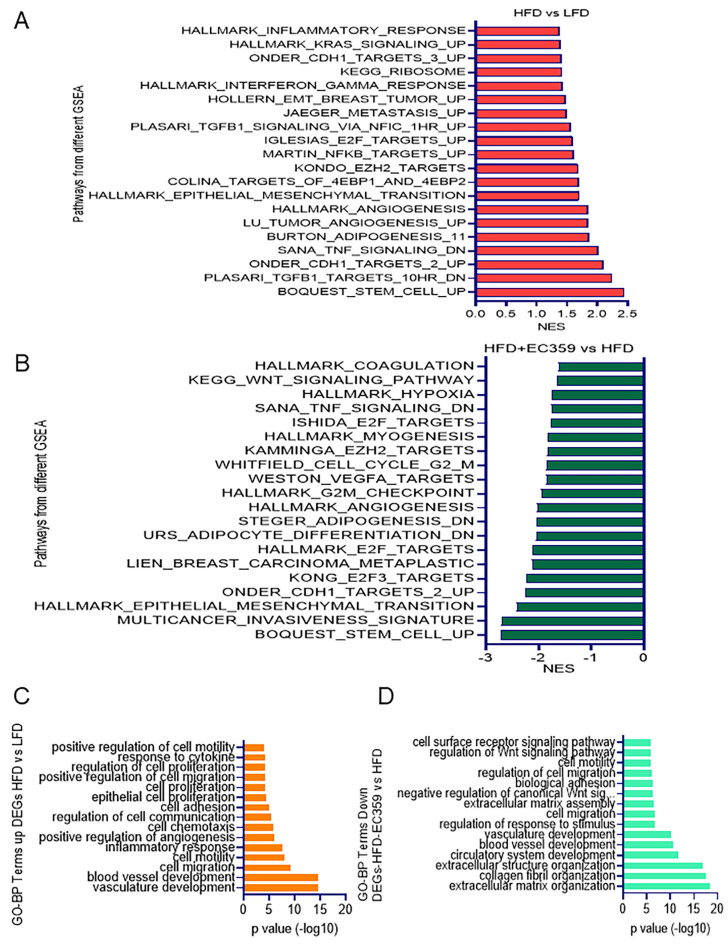
Differential gene expression analysis of TNBC syngeneic xenograft model. (**A**) GSEA reveals pathways with positive enrichment (upregulated pathways) in diet-induced obesity TNBC-4T1 syngeneic xenograft models after HFD feeding. (**B**) GSEA displays pathways with negative enrichment (downregulated pathways) in EC359-treated TNBC-4T1 syngeneic xenograft models after HFD feeding. Gene ontology analysis identifies biological processes with positive enrichment in HFD models (**C**) and negative enrichment in EC359-treated HFD models (**D**). Differentially expressed genes were filtered using log2FoldChange ≥ 2, and *p*adj < 0.01.

## Data Availability

All data generated for this study are included within this article. RNA-seq data have been deposited in the GEO database with a GEO accession number GSE280092.

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
