# Peer review of "Significance of LIF/LIFR Signaling in the Progression of Obesity-Driven Triple-Negative Breast Cancer"

_cancers, 2024, doi:10.3390/cancers16213630_

Round 1

Reviewer 1 Report

Comments and Suggestions for Authors

While this manuscript purports to reveal the significance of LIF/LIFR signaling in the progression of obesity-driven triple-negative breast cancer, the experiments described lack important controls needed to justify this conclusion. Important experimental details are also omitted. Overall, evidence for EC359 specificity is lacking - the observed effects may be ascribed to non-specific toxicity.

  1. The generation and use of adipocyte conditioned media are poorly described. Were growth factors or serum present? What was the adipocyte cell density and how long were the cells incubated before medium collection? What ratio of conditioned medium to standard medium was employed? The distinctions among ADP-CM-1 through 5 are not explained - are these different lots and if so, how were they equilibrated?

  2. Figures 1D and 1E lack an important control. The authors argue that EC359 specifically interferes with ADP-CM stimulation, but if that were the case EC359 should have no effect on control conditions. However, there is no “Control + EC359” treatment group.

  3. The Y-axis in Figure 1D is incorrectly labeled “% Cell Viability.”

  4. In Fig. 2, the authors demonstrate that ADP-CM increases STAT3 reporter activity but they do not provide direct evidence that this increase is initiated primarily through LIFR signaling.

  5. In Figs. 3 and 4, “No tumor” controls are missing. What is the evidence that EC359 is not non-selectively toxic?

Comments on the Quality of English Language

No comments

Reviewer 2 Report

Comments and Suggestions for Authors

This work studied the role of leukemia inhibitory factor receptor (LIFR) oncogenic signaling in obesity-associated TNBC and assessed the efficacy of LIFR inhibition with EC359 in blocking TNBC progression. By culturing TNBC cells with human primary adipocytes, or treated with adipocyte-conditioned medium or EC359, it showed that adipose conditions increased TNBC cell proliferation and invasion, and these effects were correlated with enhanced LIFR signaling. Accordingly, EC359 treatment reduced cell viability, colony formation, and invasion under adipose conditions and blocked adipose-mediated organoid growth and TNBC xenograft tumor growth. The findings suggest that adiposity contributes to TNBC progression via the activation of LIF/LIFR pathway, and LIFR inhibition with EC359 may represent a promising therapeutic approach for obesity associated TNBC.

Specific concerns:

1. The 1st sentence of Introduction is confusion and unclear. In addition, the cited references 1 & 2 are old, more recent ones are suggested.

2. Adipocyte-conditioned medium (ADP-CM) was used in the studies of figures 1 and 2. However, it was not clear about the labelling of ADP-CM-1, -2, -3, -5.

3. Fig. 5 showed the RNA-Seq data by analyzing 4T1-tumor xenografts. It was wondered if the same analysis was performed with human TNBC cell line MDA-MB-231-generated tumors in mice (Fig. 3).

Reviewer 3 Report

Comments and Suggestions for Authors

The manuscript by Randolph L et al., entitled: Significance of LIF/LIFR signaling in the progression of obesity-driven triple-negative breast cancer; study signaling events in breast cancer, induced by obesity conditions. Among the events addressed are: a panel of signal proteins expression prior and after adipocyte induction through assessment of RT-qPCR, Western blots exhibiting phosphorylation status by obesity induction, colony formation, Matrigel invasion, MMT for cell survival and  xenograft tumor growth, in vivo.  These are inhibited by EC359, a compound designed to  inhibit LIFR/LIF interactions. RNA-seq analyses performed on tissues of orthotopic mammary tumors versus EC-359 treated mammary tumors revealed HFD (high fat diet) induced significantly the expression of 578 genes compared to LFD (low fat diet) models. Three hundred twenty five; 325  genes were differentially expressed in the HFD + EC359 group compared to 282 the HFD group.

This is a straight forward manuscript depicting the impact of EC-325 on adipocyte induced triple negative breast cancer (TNBC).

However, a major hurdle is the fact that the authors evaluate downstream signaling events, which are commonly shared by numerous other cell surface receptors instead of showing directly the impact affecting possibly LIFR/LIF following  adipocytes treatment.

 Having said this, previous publication by this group (Viswanadhapalli  S et al., EC359-A first-in-class small molecule inhibitor for targeting oncogenic LIFR signaling in triple negative breast cancer. Mol Cancer Ther. 2019) demonstrated essentially the same pattern of signaling proteins, colony formation, Matrigel invasion and MTT for cell survival in breast cancer  MDA-MB-231 cell line. The novelty of the present manuscript relies on the induction of adipocytes induced downstream LIFR/LIF signaling events etc.

-          The manuscript is lacking significant experimental data information. For example, what is the concentration of EC325 used in the experiments? The range given in general in the method section; between 50-100 nM does not show the actual concentrations in an experiments.

-          For adipocyte CM- what is the number adipocyte cells? for how long condition medium was collected?

-          Co-culture of breast cancer cell lines and adipocytes – number of cells used?

RNA-seq analyses of 325 genes out of 578 affected by EC325 indicates the wide spectrum found affected.  It calls for further funnel narrowing down to a more specific message.

-What is the specificity / selectivity of EC-325 in breast cancer?

-The Discussion does not cover adequately the relevant literature in the field. 

     Overall, while the manuscript major aim is to study the adipocyte impact on LIFR/LIF(it is requested to show a direct impact on LIFR/LIF), and elucidate some mechanism by which adipocytes influence the receptor /pathway,  it grossly repeats a panel of signal proteins affected (as previously the authors showed; Mol Cancer Ther. 2019). The added value information here is limited.

Comments on the Quality of English Language

The manuscript reads well and fluently.

Round 2

Reviewer 3 Report

Comments and Suggestions for Authors

Accept as is in its current form.